# Ultrasound Assisted Extraction of Saponins from *Hedera helix* L. and an In Vitro Biocompatibility Evaluation of the Extracts

**DOI:** 10.3390/ph15101197

**Published:** 2022-09-28

**Authors:** Adina I. Gavrila, Rodica Tatia, Ana-Maria Seciu-Grama, Isabela Tarcomnicu, Cristina Negrea, Ioan Calinescu, Christina Zalaru, Lucia Moldovan, Anca D. Raiciu, Ioana Popa

**Affiliations:** 1Department of Bioresources and Polymer Science, Faculty of Chemical Engineering and Biotechnologies, University Politehnica of Bucharest, 011061 Bucharest, Romania; 2Department of Cellular and Molecular Biology, National Institute of Research and Development for Biological Sciences, 060031 Bucharest, Romania; 3Cytogenomic Medical Laboratory, 014462 Bucharest, Romania; 4Department of Organic Chemistry, Biochemistry and Catalysis, Faculty of Chemistry, University of Bucharest, 050663 Bucharest, Romania; 5SC HOFIGAL SA, 042124 Bucharest, Romania

**Keywords:** *Hedera helix*, saponins, ultrasound, cytotoxicity, biocompatibility

## Abstract

The aim of this study was to establish the best ultrasound assisted extraction (UAE) conditions of saponins from *Hedera helix* L. leaves and to evaluate the in vitro biocompatibility of the extracts richest in saponins. Different parameters, such as extraction time, temperature, ultrasound power, solvent to plant material ratio, and solvent concentration, were investigated. The most efficient extraction conditions were a temperature of 50 °C, an ultrasound amplitude of 40%, an extraction time of 60 min, a plant material to solvent ratio of 1:20 (w:v), and 80% ethanol as solvent. In vitro cytotoxicity of the extracts richest in saponins and their influence on the DNA content of L929 (NCTC) fibroblasts were tested. Until 200 µg/mL, the studied extracts were cytocompatible with L929 fibroblast cell lines at 48 h of treatment. These in vitro cell culture results provide useful information for further applications of *Hedera helix* extracts in a pharmaceutical field.

## 1. Introduction

Common ivy (*Hedera helix* L.), which belongs to *Araliaceae* family, is a perennial evergreen woody liana that can be found in Southern, Western, and Central Europe, Asia, and North America [1]. *Hedera helix* is rich in triterpene saponins, such as hederacoside C, hederacoside D, hederacoside B, α-hederin, and β-hederin. It also contains polyphenolic compounds (phenolic acids, flavonoids, coumarins, anthocyanins etc.), vitamins, steroids, amino acids, volatile oils, polyacetylenes, and β-lectins [1]. The main component from *Hedera helix* leaves is the triterpenoid saponin hederacoside C which, along with α-hederin and aglycon hederagenin, are considered to be phytocompounds with important biologic activity. These compounds are responsible for the antimicrobial [2], anti-inflammatory [3], antiviral [4], antiparasitic [5], antioxidant [6], antifungal [7], antimutagenic [8], cytotoxic [9], and immunological [10] activities of *Hedera helix*.

Considering the beneficial effect of the bioactive compounds, different extraction methods to obtain extracts rich in saponins were investigated. There are reported, in literature, various conventional methods, such as: maceration and Soxhlet or heat reflux extractions. However, these techniques present low efficiency and selectivity due to high temperatures, long extraction times, high energy consumption, and high quantities of solvent used [11]. To overcome these drawbacks, various non-conventional methods to extract saponins were successfully employed. Li et al. reported that the extraction of saponins from *Aralia taibaiensis* root bark by ultrasounds (at a temperature of 61 °C, an ethanol concentration of 73%, a solid to liquid ratio of 16 g/mL, and an extraction time of 34 min) led to better yields as compared with the conventional hot reflux extraction [12]. Akbari et al. studied the influence of different parameters on the microwave assisted extraction of saponins from fenugreek seed, concluding that the concentration of ethanol was the most significant factor, while temperature exhibited the least contribution on the total saponins content [13]. Supercritical fluid extraction of saponins from *Medicago sativa* was successfully employed by Kielbasa et al. Moreover, they developed a method for qualitative and quantitative analysis of saponins that involved acid hydrolysis with concentrated hydrochloric acid of the extracts and purification by supercritical fluid extraction with C18 [14]. Another study presents the subcritical-water extraction of triterpene saponins from *Orostachys japonicus*. Different temperatures and extraction times were investigated, concluding that the triterpene saponins can be observed only at 220 °C and 15 min of extraction [15]. Shahi et al. studied the pulsed electric field extraction of saponins from Chubak root and the possibility to use the extracts as emulsion and foam system for ice cream. They revealed that increasing the pulse number from 10 to 65 and the voltage from 1 to 4 kV/cm led to a better yield. This resulted in an increase of foaming ability and emulsion stability [16]. The enzymatic assisted extraction is another interesting technique being employed successfully to extract saponins from *Sapindus mukorossi*. Chen et al. reported a total saponin content of approximatively 116 mg/mL for 3 h of extraction, a plant material to solvent ratio of 1:5, and a cellulase concentration of 0.55% [17].

Ultrasound assisted extraction (UAE) is a promising technique to intensify the extraction process of bioactive compounds from vegetal materials, being environmentally friendly. In heterogenous mixtures, such as solid-liquid extraction of bioactive compounds from herbs, the cavitation phenomenon can promote the disruption of cell walls which results in an increase of the mass transfer rate. This means that the cavitation bubbles generated near the surface of the plant material will collapse and act as a microjet, which will damage the cellular tissue and, implicitly, dislodge the targeted compounds [18]. The UAE can be performed using two types of equipment: ultrasonic baths and horns. In baths, the ultrasound energy is unevenly distributed, and it is of low intensity. On the contrary, the sonication process for horns is more efficient due to a high localized intensity of ultrasonic energy [19]. Thus, to intensify the extraction process, the use of an ultrasound horn can be more suitable.

The purpose of this study was to evaluate the process parameters of UAE of saponins from *Hedera helix* in order to establish an extraction protocol for the major saponins found in *Hedera helix* leaves. The influence of the process main factors: ultrasonic power, temperature, plant to solvent ratio, and extraction solvent (ethanol concentration) on the saponin content of *Hedera helix* leaves extracts was studied. Subsequently, in vitro analyses of the extracts, richest in saponins, were performed in order to determine their cytotoxicity and the influence on the DNA content of L929 fibroblasts. To the best of our knowledge, a protocol for the UAE of saponins from *Hedera helix* has been evaluated, for the first time, in this study.

## 2. Results and Discussion

### 2.1. Influence of UAE Parameters on the Extraction Efficiency

In order to establish the best extraction conditions, the study of the various parameters influence on the extraction process is of great importance. The main components from *Hedera helix* extracts, hederacoside C, α-hederin, and the aglycon hederagenin, are considered to be the active substances and were quantified by high-performance liquid chromatography (HPLC) analysis.

#### 2.1.1. The Influence of Ultrasound Power

In the study of ultrasound assisted extractions, the acoustic power is an important parameter since it indicates the quantity of energy that is effectively transferred to the liquid. Literature data show that a high ultrasonic power can degrade the vegetal material by inducing greater shear forces and turbulences [20]. For this reason, the influence of acoustic power on the extraction process is evaluated in order to find the power value that provides the best extraction results.

Due to some interference with other compounds, the spectrophotometric method is not specific enough for saponins. It is important to note that the total saponin content (TSC) was performed only for the purpose of comparing the -extracts. More specifically, chromatographic methods are required to accurately quantify the main saponins from the extracts. In any case, given the fact that this spectrophotometric method is the most used assay of TSC screening, it can be considered valid to compare the extraction conditions [21].

The main saponins analyzed by HPLC and the TSC variation with the sonication power and the ultrasound amplitude are shown in Figure 1. The sonication power might promote the dissipation of the mechanical energy producing a “heating effect” at the surface of the plant material and therefore increasing the diffusivity effects. Likewise, the formation of acoustic cavities and microchannels are considered the main reasons for the intensification of the mass transfer phenomena in ultrasound assisted extractions [22].

As shown in Figure 1A, the lowest content is obtained for the extraction of saponins by conventional method (38 mg DE/g DM). The highest TSC (45 mg DE/g DM) is achieved by UAE, for an ultrasonic power of 27.9 W. These results show that the ultrasonic extraction technique is very efficient in the extraction of saponins as compared with the conventional method. The one-way ANOVA analysis demonstrated that by increasing the ultrasonic power from 0 to 27.9 W the TSC increases significantly (*p* < 0.05). If the power is increased to 58.9 W, the TSC variation is not significant. Thus, the following experiments were performed at an amplitude of 40%, instead of 60%.

The ethanolic extracts were also analyzed by HPLC in order to determine the saponins content (hederacoside C, α-hederin, and hederagenin—Figure 1B). As shown in Figure 1B, hederacoside C is the saponin extracted in the highest amount, followed by α-hederin. Increasing the ultrasonic power leads to an increase in the quantity of extracted saponins.

#### 2.1.2. The Influence of Temperature and Extraction Time

Temperature plays a particularly important role in saponin extraction. In order to prevent the degradation of thermolabile compounds, the extraction temperature must be chosen according to the targeted extraction components [23]. An increase of temperature promotes compounds solubility, reduces viscosity and surface tension of the extraction medium, facilitates molecular collisions at the mixture interface, increases mass transfer, and induces an increase of vapor pressure [24]. The latter leads to the cushioning effect of vapors contained inside the bubbles during cavitation, meaning that the intensity of cavitation is inversely proportional with temperature [25,26].

The amount of extracted saponins at different temperatures (30, 40, and 50 °C) is shown in Figure 2. At a temperature of 30 °C, a maximum content of approximately 42.69 mg DE/g DM is obtained. At low temperatures, the number of bubbles generated is small, but they collapse violently, increasing the mass transfer between vegetal material and solvent [27]. Increasing the temperature with 10 °C, the saponin content also increases, reaching a maximum value of 48.52 mg DE/g DM. At a temperature of 50 °C, the use of acoustics leads to a higher amount of extracted saponins (50.27 mg DE/g DM). The increase of temperature leads to a significant increase of TSC (*p* < 0.05) for all extraction times studied.

The extraction time is another important parameter that influences the extraction yield. The content of extracted bioactive components increases with time until an equilibrium between the constituents inside of the plant cells and the ones already solubilized by the solvent is reached [28]. Figure 2 shows the extraction results obtained for different times. The saponin amount increased quickly in the first 20 min. In addition, it can be noticed in Figure 2 that, at a temperature of 50 °C, the saponin content increases up to 80 min, after which a slight decrease occurs. The same behavior is observed at a temperature of 40 °C. These results can be explained by the degradation of the thermolabile components at both 40 and 50 °C. Increasing the extraction time, all the plant cells will be completely broken by acoustic cavitation effect, and the extraction yield will increase [29]. The dissolved constituents will also re-adsorb on the crushed vegetal material particles, due to their relatively large specific surface areas, reducing the quantity of the recovered compounds. Thus, a longer extraction time after the maximum extraction yield is achieved is useless [30]. The ANOVA analysis along with multiple comparison post hoc tests (n = 21) showed that TSC increases significantly up to 60 min (*p* < 0.05). Given these results and also from an economic point of view (higher energy consumption for a longer time), the time of 60 min is considered to be optimum.

The content of bioactive components in the extracts was also evaluated by HPLC analysis. For all three temperatures used, only the extracts obtained at 60 min were analyzed. The results are shown in Figure 3A. However, since the highest TSC values were obtained at 50 °C and in order to assess the stability of key compounds over time, the main saponins, at this temperature, were evaluated for all extraction times (see Figure 3B).

In Figure 3, it can be noticed that hederacoside C is directly proportional to extraction time and temperature. However, α-hederin and hederagenin contents remain almost constant regardless of these two parameters. According to the HPLC analysis, these components are extracted from the first minutes of extraction (see Figure 3B). Hederacoside C content increases up to 100 min, while α-hederin and hederagenin amounts remain almost constant. This could be explained by a slight lability of the key compounds at long extraction times or that the extraction was completed.

#### 2.1.3. The Influence of Plant Material to Solvent Ratio

Solid to liquid ratio is a fundamental variable that can considerably affect the saponin extraction efficiency in the UAE process. The ANOVA analysis along with multiple comparison post hoc tests (n = 3) showed that TSC increases significantly (*p* < 0.05) by increasing the plant material to solvent ratio from 1:10 to 1:20. As shown in Figure 4A, at a plant material to solvent ratio of 1:10 (w:v), the TSC was 46.95 mg DE/g DM, while at a ratio of 1:20 the TSC was 51.27 mg DE/g DM.

The low saponin content at a plant material to solvent ratio of 1:10 is due to the too concentrated vegetal material, which reduces the transfer of ultrasonic energy [20]. Generally, a larger solvent volume can dissolve more effectively the active components, leading to an enhancement of the extraction yield [31]. However, increasing the plant material to solvent ratio from 1:20 to 1:30 (w:v), the increase in TSC was not significant (post hoc *t*-test *p* > 0.05). Moreover, it is very important to diminish the consumption of plant material and solvent, while prioritizing the highest extraction yields of the targeted active compounds. Considering the above, the plant material to solvent ratio of 1:20 (w:v) was chosen for the further experiments.

#### 2.1.4. The Influence of Ethanol Concentration

Another important parameter for saponins extraction is the selection of solvent and the influence of the solvent concentration. In order to choose the right solvent for UAE, different properties, such as solubility of the active components, viscosity, surface tension, and vapor pressure of the solvent, are important. These properties will affect the acoustic cavitation and also the cavitation threshold [32]. The initiation of cavitation phenomenon in a solvent requires that the pressure during the rarefaction cycle must overcome the cohesive forces between the liquid molecules. A solvent with a low vapor pressure is favored because the collapse of the cavitation bubble is more intense as compared with solvents with a high vapor pressure [33]. Even if it is potentially explosive and inflammable, ethanol is widely used for extraction due to its low toxicity, availability at high purity, and low price, as well as being considered a green solvent [34].

Analyzing the experimental data shown in Figure 5A, it can be noticed that the highest TSC was obtained when 80% ethanol as solvent was used (56.87 mg DE/g DM). The one-way ANOVA analysis along with multiple comparison post hoc test (n = 15) showed that by increasing the ethanol concentration from 20 to 80% the TSC increased significantly (*p* < 0.05). When the solvent contacts the plant material, it will surround it and a slow swelling process will begin. Hydrogen bonds will be developed between solvent (ethanol) and the hydroxyl groups from the cellulosic structure of the plant material, and this superficial layer of solvent will hamper the extraction efficiency by blocking the diffusion process. This surrounding layer is rich in extracted compounds from the plant. Using ultrasound, the hydrogen bonds are broken and the superficial layer is continuously removed due to the high-speed jets produced by the collapse of the asymmetrical bubbles, and the diffusion process is enhanced [35].

The extracts obtained for a concentration of 80% ethanol were analyzed by HPLC and compared with those when 96% ethanol was used. The results are shown in Figure 5B. The highest content of hederacoside C was also extracted at an ethanol concentration of 80%.

### 2.2. In Vitro Biocompatibility Testing

To explore the potential application of the 96% and 80% ethanolic extracts in the pharmaceutical field, their in vitro biocompatibility was evaluated. Moreover, the high content in hederacoside C of these two extracts contributed to the selection of both extracts in order to assess their cytotoxicity and the influence of plant extracts on the DNA content.

#### 2.2.1. Cytotoxicity of Extract Samples

The cell viability was quantified by MTT test, which is an indicator of living cells metabolism, especially of mitochondrial enzyme activity. The cytotoxicity of the 96% and 80% ethanolic *Hedera helix* extracts was compared to that of hederacoside C, as saponin standard. For a concentration range from 10 to 200 μg/mL, the MTT results showed that the percentage of viable cells, after 24 h of treatment with studied samples, varied between 108.95 and 119.34% in the 96% ethanolic extract and between 99.89 and 118.16% in the 80% ethanolic extract. This indicates that both tested samples were not cytotoxic up to a concentration of 200 μg/mL, according to ISO 10993-5 (see Figure 6). Moreover, the values were significantly higher (*p* < 0.05) than those of untreated cells (control). However, for a concentration of 400 μg/mL, the 80% ethanolic extract manifested cytotoxicity when a cell viability of only 76.5% was obtained.

Statistic examination after 48 h of treatment of the cells in the presence of studied samples showed that the viability of the L929 fibroblasts was similar for a concentration range from 10 to 50 μg/mL. As shown in Figure 7, for the interval of 100–200 μg/mL, the viability values were significantly higher (*p* < 0.05) in the case of 80% ethanolic extract as compared with 96% ethanolic extract (viability results of 101.7–107.82% and 94.9–97.9%, respectively, after 48 h of treatment). However, at 400 μg/mL, the 80% ethanolic extract manifested severe cytotoxic effect with a significant reducing of the cell viability to only 45.09%, while the 96% ethanolic extract was slightly cytotoxic (78.80%).

Hederacoside C at both 24 and 48 h of treatment was biocompatible at a concentration up to 200 μg/mL, while it exerted low cytotoxicity (71.03% viability) against fibroblasts cells at 400 μg/mL.

#### 2.2.2. Influence of Plant Extracts on the DNA Content

L929 fibroblast cells were incubated in the presence of different concentrations of samples for two days, in standard conditions of cultivation. The degree of cell proliferation was fluorometrically assayed in cell lysate using DNA quantification. The obtained results are shown in Table 1 and indicate that the interaction of the samples with L929 fibroblast cells was influenced by sample concentration.

The results showed that throughout the culture period the DNA content was increased at a concentration of 100 µg/mL for all tested samples as compared with the control (untreated cells). A high DNA content (above 90%) was observed for the cells treated with tested samples at a concentration of 200 µg/mL. At a concentration of 400 µg/mL, the total DNA content ranged from 50.84% for hederacoside C to 70.46% for the 80% ethanolic extract and to 75.69% for the 96% ethanolic extract.

## 3. Materials and Methods

### 3.1. Materials

*Hedera helix* leaves were harvested in the summer of 2021 at Hofigal S.A., Romania. The fresh leaves were dried in an air flow-heating oven at 60 °C to a constant weight (the final moisture content was 5.8%). The dried leaves were then ground using an electric grinder and sieved to a particle size under 315 μm. The ground leaves were dosed in samples of 25 g (in a sealed plastic vessel) and stored at 4–5 °C until they were used for extraction of the bioactive components.

The standard used for saponin determination was diosgenin, purchased from Sigma-Aldrich (Bucharest, Romania). The solvent ethanol, vanillin and sulfuric acid were of analytical grade and were purchased from Merck. For the HPLC quantification of saponins, the following standards were used: α-hederin, hederacoside C, and hederagenin (Sigma-Aldrich, Bucharest, Romania). Methanol, acetonitrile, water (all for chromatography, ultragradient grade), and formic acid (for analysis, 85%) were purchased from Carlo Erba (Cornaredo, Italy).

### 3.2. Saponins Extraction Procedure

The saponins extraction was carried out in batch system using a special jacketed glass reactor. In order to maintain the extraction temperature at the proposed value, circulating water at appropriate temperature was introduced into the jacket during all extractions. For the ultrasound extraction, a Vibracell VCX750 (Sonics&Materials, Inc.; Newtown, CT, USA) ultrasonic probe was inserted into the extraction reactor, directly in the mixture. The ultrasound amplitude was set at 20, 40, and 60%, which correspond to 5.5, 27.9, and 58.9 W, respectively. The *Hedera helix* leaves were submitted to extraction at different temperatures (30, 40, and 50 °C) for 1, 5, 10, 20, 40, 60, 80, and 100 min. The plant to solvent ratio was varied as follow: 1:10, 1:20, and 1:30 (w:v). The solvent used was a mixture of 0, 20, 40, 60, 80, and 96% ethanol in water. For the conventional extraction procedure, a stirring system was used instead of an ultrasonic probe (900 rpm). After the extractions, the solid plant material was separated by centrifugation for 5 min at 2500 rpm. The supernatant was evaporated to dryness on a rotavapor system and the residue is kept to freezer until HPLC analysis. All experiments were performed in triplicate.

### 3.3. Analysis

#### 3.3.1. Total Saponins Content Determination

The TSC was colorimetrically evaluated using the method reported by Hiai et al., with minor modifications [36]. This method is based on the reaction between C-3 carbon of saponins with vanillin and sulfuric acid, to produce chromophores absorbing at 544 nm. As reference substance diosgenin was used [37]. From the fresh extracts, 0.5 mL were taken and evaporated to dryness on a rotavapor. The residue was dissolved in 10 mL of 80% methyl alcohol aqueous solution. A volume of 0.5 mL of the methanolic solution was mixed with 0.5 mL of 8% vanillin solution in ethanol and 5 mL of 72% sulfuric acid solution. The vial containing this mixture was placed in a water bath at 60 °C for 10 min, then cooled in a water-ice mixture for 4 min. The absorbance was measured at 544 nm using a Shimadzu UV mini-1240 UV/vis Scanning Spectrophotometer, 115 VAC (Duisburg, Germany). The results were quantified as milligrams of diosgenin equivalents per 1 g of dry matter (mg DE/g DM) using a standard curve corresponding to 40–550 mg/L diosgenin solution. Each analysis was performed in triplicate.

#### 3.3.2. HPLC-MS/MS Analysis of Major Saponins

The targeted compounds quantification was carried out by liquid chromatography coupled with tandem mass spectrometry (LC-MS/MS). The analyses were performed on a triple quadrupole mass spectrometer model API3200 (Sciex, Toronto, Canada) coupled with an Infinity 1260 binary pump and autosampler (Agilent, Santa Clara, CA, USA). Analyst software version 1.5.2 was used for data acquisition and processing. The samples were separated on a Luna amino (NH_2_) column (150 × 2 mm, 3 μm, 100 Å), produced by Phenomenex (Torrance, CA, USA), using a mobile phase composed of water (A) and acetonitrile (B), both containing as modifier 0.1% formic acid. The separation was achieved in hydrophilic interaction liquid chromatography mode (HILIC). The flow rate was 0.3 mL/min and the gradient started at 5% A (maintained for 0.5 min). The aqueous phase was first increased to 50% in 0.5 min and kept for another 0.5 min; then it was increased again to 95% in 0.5 min. The 95% aqueous composition was maintained for 2 min, followed by column equilibration to initial conditions. Hederagenin, α-hederin and hederacoside C eluted at 1.41, 3.99 and 3.97 min, respectively.

The dried extracts were dissolved in 1 mL of methanol by thorough vortexing (5 min) with an IKA Vortex 3 (Staufen, Germany). Further, it was diluted—1:10,000 (v:v)—with methanol before injecting 10 µL of the solution in the LC-MS/MS system. The mass spectrometer ESI interface was operated in negative ion mode. Acquisitions were carried out in multiple reaction monitoring mode using the specific traces of each saponin. The quantification was performed on calibration curves prepared with different concentrations of saponin standards in methanol, each adjusted to match the concentrations in the extracts. Thus, the calibration curve ranged from 15 to 1500 ng/mL for hederacoside C, from 5 to 500 ng/mL for α-hederin, and from 2.5 to 250 ng/mL for hederagenin. This specific range of dilutions was the most appropriate to achieve linearity and to reduce the matrix effects of the co-extracted components.

#### 3.3.3. Measurement of Cytotoxicity

Mouse fibroblasts from NCTC clone L929 cell line were grown in minimum essential medium supplemented with 10% fetal calf serum, 2 mM L-glutamine, and 1% mixture of antibiotics. The culture was maintained in an incubator at 37 °C, in humid atmosphere with 5% CO_2_. Cell suspension was seeded at a density of 4 × 10⁴ cells/mL in culture plates and incubated in standard conditions for 24 h to allow cell adhesion. Then, the saponins-rich extracts performed with 96 and 80% ethanol were added in the culture medium at concentration values of 10, 50, 100, 200, and 400 μg/mL. The plates were incubated at 37 °C for 24 h and 48 h, respectively.

#### 3.3.4. MTT Assay

Cytotoxicity of treated cells with extract samples was analyzed using the MTT method in which the tetrazolium salt reduces the enzyme mitochondrial dehydrogenase in metabolically active cells, as previously described by Mosmann et al. [38]. Briefly, after the treatment, the sample solutions were replaced with fresh medium containing MTT solution, in a 10:1 (v:v) ratio, and the plates were incubated at 37 °C for 3 h. Then, 100 μL of isopropanol was added to each well and stirred for 15 min to dissolve the formazan crystals. The absorbance of colored solutions was measured at 570 nm, using a Berthold Mithras microplate reader (Germany). The measured absorbance is directly proportional to cell viability and the results were calculated using the following equation:Cell viability (%) = Sample absorbance/Control absorbance × 100.(1)

The control was untreated cells and the cells treated with 0.003% hydrogen peroxide served as a positive control. Hederacoside C in the same concentrations with the extract samples was used as a standard saponin. Three separate experiments were conducted, and the results were expressed as mean ± SD.

#### 3.3.5. Statistical Analysis

All measurements were carried out in triplicate and the data were expressed as mean value ± SD (standard deviation) for triplicate of samples (n = 3). All the results were subjected to one-way analysis of variance (ANOVA). Statistical analysis of the data was performed using the multiple comparison post hoc *t*-tests in order to detect the significant statistical differences between the averages of the TSC of two or more independent groups. The differences were considered statistically significant at *p* < 0.05 as a minimal level of significance.

#### 3.3.6. Quantification of DNA Cellular Content

DNA content of L929 fibroblast cells was fluorometrically assayed, as previously described by Rapa et al. [39]. L929 fibroblast cells were incubated in the presence of different concentrations of samples in standard conditions of cultivation. After 48 h, the cells were washed with phosphate buffered saline and incubated in cell lysis buffer (0.5 mL) containing 30 mM saline-sodium citrate (Sigma-Aldrich, Bucharest, Romania) and 0.2 mg/mL sodium dodecyl sulphate at 37 °C for 1 h. The released DNA content was evaluated with a Quant-iT dsDNA HS Assay kit (Invitrogen, Waltham, MA, USA) using Qubit fluorometer (Invitrogen, USA), according to the manufacturer’s instructions. The results were reported as a percentage of DNA content relative to the control cell culture (untreated cells) considered to be 100%.

## 4. Conclusions

The aim of this study was to develop an efficient method of extracting saponins from *Hedera helix* leaves by UAE in batch mode. This technique was evaluated as a simpler and more effective alternative to conventional extraction method for the isolation of saponins. The *Hedera helix* leaves samples were extracted by direct sonication using an ultrasound probe horn. To determine the factors influencing the extraction process, different parameters, such as extraction temperature and time, sonication power, plant material to solvent ratio, and ethanol concentration, were studied. Higher TSC and higher amounts of hederacoside C and α-hederin were achieved at a temperature of 50 °C within 60 min, using an ultrasound amplitude of 40%, a plant material to solvent ratio of 1:20 (w:v) and an ethanol concentration of 80%.

Another purpose of this study was to evaluate the in vitro biocompatibility of *Hedera helix* leaves extracts and their influence on the DNA content of L929 fibroblasts. For these analyses, the extracts obtained under the optimal extraction conditions were used. The cytotoxicity assay of the samples indicated that until 200 μg/mL both ethanolic extracts were biocompatible on L929 fibroblast cells. The DNA content of fibroblasts treated with the extract samples and with the saponin standard were in accordance with the cell viability results determined by MTT assay. Both methods showed that the viability values of cells cultivated in the presence of studied extracts were dependent on the tested samples concentration. Additional in vitro and in vivo studies are required to characterize *Hedera helix* extracts in order to be used in pharmacological applications.

## Figures and Tables

**Figure 1 pharmaceuticals-15-01197-f001:**
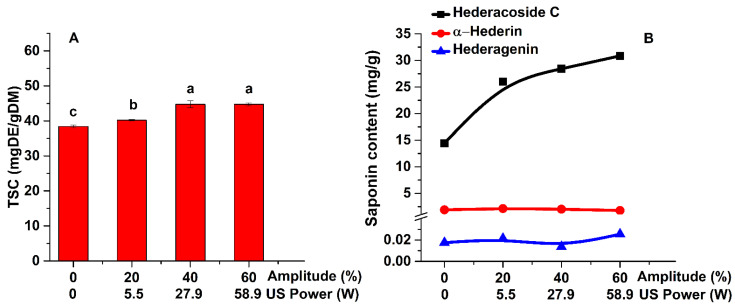
Influence of ultrasonic power on the TSC (**A**) and on the main saponins (**B**) of *Hedera helix* leaves extracts (extraction conditions: a temperature of 30 °C, an extraction time of 60 min, a plant material to solvent ratio of 1:20 (w:v), continuous sonication, an ethanol concentration of 96%). Data were analyzed using one-way ANOVA (*p* < 0.05) and multiple comparison post hoc *t*-tests (n = 6). Different letters (a–c) within graph shows the significant difference between groups.

**Figure 2 pharmaceuticals-15-01197-f002:**
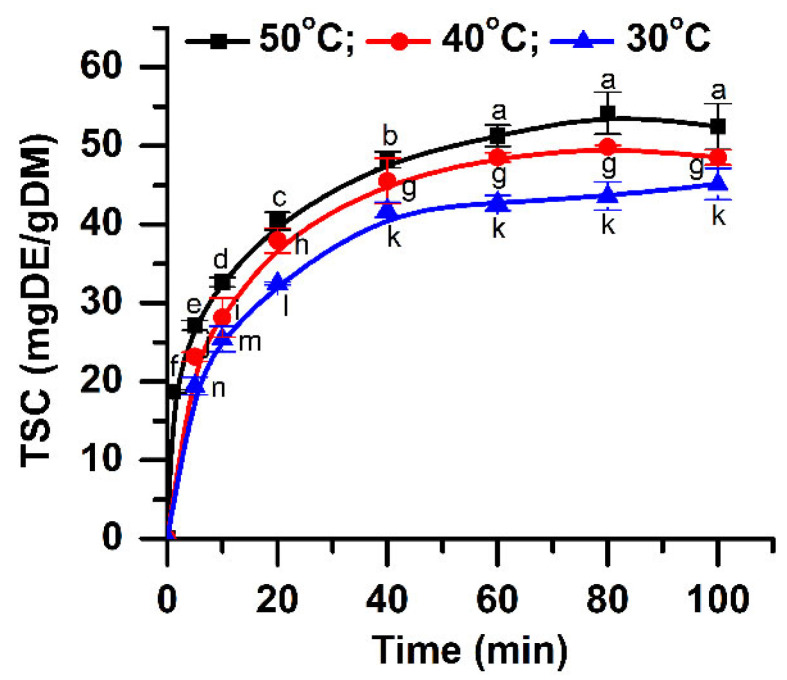
Influence of extraction time and temperature on the TSC (extraction conditions: a plant material to solvent ratio of 1:20 (w:v), continuous sonication, an ethanol concentration of 96%). Data were analyzed using one-way ANOVA (*p* < 0.05). Different letters (a–n) within graph shows the significant difference between groups.

**Figure 3 pharmaceuticals-15-01197-f003:**
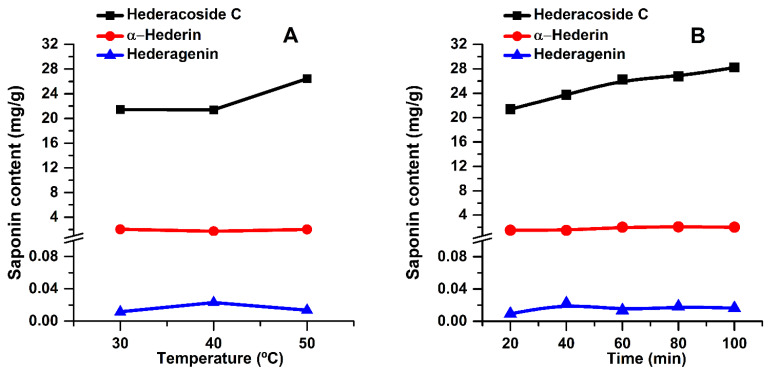
Influence of temperature (**A**) and extraction time (**B**) on the main saponins of the extracts (extraction conditions: a plant material to solvent ratio of 1:20 (w:v), continuous sonication, an ethanol concentration of 96%).

**Figure 4 pharmaceuticals-15-01197-f004:**
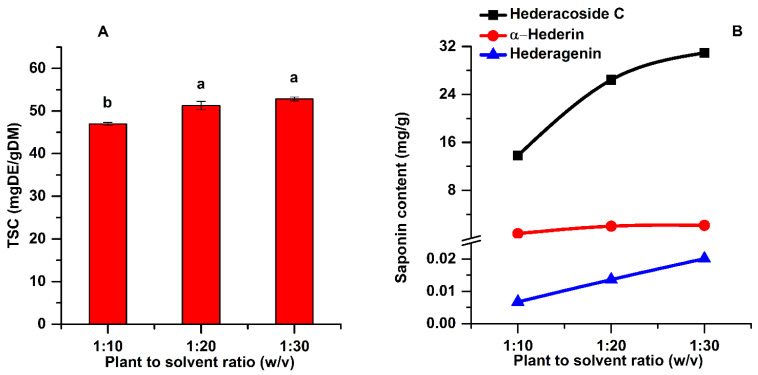
Influence of plant material to solvent ratio on the TSC (**A**) and on the main saponins (**B**) of the extracts (extraction conditions: a temperature of 50 °C, an extraction time of 60 min, continuous sonication, an ethanol concentration of 96%). Data were analyzed using one-way ANOVA (*p* < 0.05) and multiple comparison post hoc *t*-tests (n = 3). Different letters (a,b) within graph shows the significant difference between groups.

**Figure 5 pharmaceuticals-15-01197-f005:**
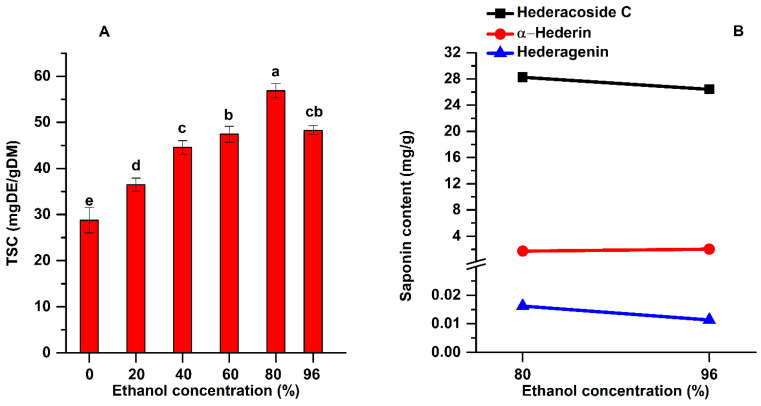
Influence of ethanol concentration on the TSC (**A**) and on the main saponins (**B**) of the extracts (extraction conditions: a temperature of 50 °C, an extraction time of 60 min, continuous sonication, a plant to solvent ratio of 1:20 (w:v). Data were analyzed using one-way ANOVA (*p* < 0.05) and multiple comparison post hoc *t*-tests (n = 15). Different letters (a–e) within graph shows the significant difference between groups.

**Figure 6 pharmaceuticals-15-01197-f006:**
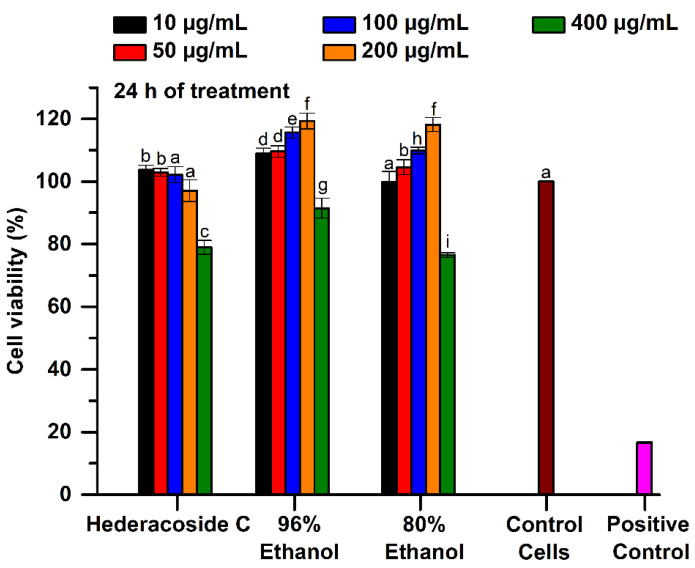
Cell viability of L929 fibroblasts cultured in the presence of 96% and 80% ethanolic *Hedera helix* extracts and hederacoside C as saponin standard, assessed by MTT assay, at 24 h of treatment. Untreated cells in the culture medium and the cells cultivated in the presence of 0.003% H_2_O_2_ were used as controls. Different letters (a–i) shows the significant difference between samples (ANOVA, *p* < 0.05).

**Figure 7 pharmaceuticals-15-01197-f007:**
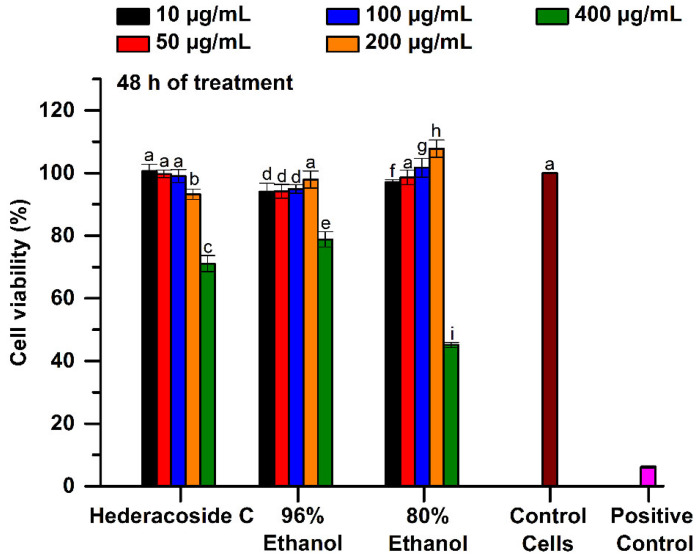
Cell viability of L929 fibroblasts cultured in the presence of 96% and 80% ethanolic *Hedera helix* extracts and hederacoside C as saponin standard, assessed by MTT assay, at 48 h of treatment. Untreated cells in the culture medium and the cells cultivated in the presence of 0.003% H_2_O_2_ were used as controls. Different letters (a–i) shows the significant difference between samples (ANOVA, *p* < 0.05).

**Table 1 pharmaceuticals-15-01197-t001:** DNA content of L929 fibroblast cells cultivated in the presence of samples. Different letters (a–h) shows the significant difference between samples (ANOVA, *p* < 0.05).

Sample	100 µg/mL	200 µg/mL	400 µg/mL
96% ethanolic *Hedera helix* extract	103.07 ± 0.67 ^c^	94.15 ± 2.00 ^e^	75.69 ± 0.75 ^f^
80% ethanolic *Hedera helix* extract	101.23 ± 3.78 ^d^	93.23 ± 1.57 ^e^	70.46 ± 1.51 ^g^
hederacoside C	105.06 ± 0.79 ^b^	100.61 ± 0.74 ^a^	50.84 ± 1.24 ^h^
control	100.00 ± 0.6 ^a^	100.00 ± 0.60 ^a^	100.00 ± 0.60 ^a^

## Data Availability

Data is contained within the article.

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
