# Peer review of "Ultrasound Assisted Extraction of Saponins from Hedera helix L. and an In Vitro Biocompatibility Evaluation of the Extracts"

_pharmaceuticals, 2022, doi:10.3390/ph15101197_

Round 1

Reviewer 1 Report

In this study, the research evaluated the process parameters of UAE of saponins from Hedera helix L. in order to establish an extraction protocol for the major hederasapo nins found in ivy leaves. Subsequently, in vitro analyses of the extracts, richest in saponins, were performed in order to determine their cytotoxicity and the influence on the DNA content of L929 fibroblasts

Overall, the manuscript is technically sound and the research ideas appears justified. Then this paper should be accepted.

Author Response

In this study, the research evaluated the process parameters of UAE of saponins from Hedera helix L. in order to establish an extraction protocol for the major hederasaponins found in ivy leaves. Subsequently, in vitro analyses of the extracts, richest in saponins, were performed in order to determine their cytotoxicity and the influence on the DNA content of L929 fibroblasts.

Overall, the manuscript is technically sound and the research ideas appears justified. Then this paper should be accepted.

Response: Thank you for your comments on the manuscript and for appreciating our work.

Reviewer 2 Report

The manuscript entitled ‘Ultrasound Assisted Extraction of Saponins from Hedera Helix L. and in vitro Biocompatibility Evaluation of the Extracts’ is a manuscript on extraction methods useful to obtain Hedera helix extracts of leaves with high amount of saponins.

The same authors have already published an abstract on the same topic: “Methods of Obtaining Extracts from Hedera helix L. Leaves and Evaluation of the Total Saponins Content” in Chem. Proc. 2022, 7, 56. https://doi.org/10.3390/chemproc2022007056. Some patents are already available (PI 0414535-6 BI) by other inventors.

In scientific manuscripts, the name of plants must be written as binomial scientific name and not using the common name. Thus, ‘Hedera helix extracts’ should be used, not ivy extracts.

Critical points

-          The binomial scientific name of plants must always be written in italics, the first name with the first letter uppercase and then lowercase as the second term, such as Hedera helix

-           The last sentence of Abstract (L.22-24) is not clear; What is the meaning? The sentence should be modified to clear the sense of biological experiments.

-          The authors should improve the introduction and, in general, the text. It is very nonspecific... A description of other methods of extraction published in the literature must be reported.

-          Results: line 108. Are the authors sure about ‘by the conventional method (34 mgDE / gDM)’? From figure 1a, the value appears higher… > 35 mg DE/g DM.

-          Results L.137-251. The description and discussion of the experimental conditions used for saponin extraction are too extensive and repetitive. This part must be written in shorter form.

-          Table 1 What is the statistical significance of the reported values? There is probably no statistical difference among the results obtained with 100 and 200 µg/mL. A statistical test must be performed. Furthermore, the data in the table are different from those reported in the text. 

-          The conclusion must be improved. From the reported results, there is no significant variation of cell proliferation and therefore DNA content up to 200 µg/ml of extracts, while 400 µg/mL are cytotoxic. Overall, one-way analysis of variance (ANOVA) and an appropriate post hoc test must be done.

-          What is the relation between the L929 fibroblast line and the immune stimulating activity of Hedera helix extracts? Please, this point must to be explained and/or the objective of the section in the manuscript “UAE of Active Principles from Ivy Leaves - Selection of Extraction Conditions in Correlation 70 with the Immunostimulatory Compounds from the Extracts” must be changed.

-  There are several typographical errors... such as ‘in vitro’ should be written in italics, etc.

 In the pdf of the manuscript, some critical points/words are highlighted in  orange colour.

Author Response

The manuscript entitled ‘Ultrasound Assisted Extraction of Saponins from Hedera Helix L. and in vitro Biocompatibility Evaluation of the Extracts’ is a manuscript on extraction methods useful to obtain Hedera helix extracts of leaves with high amount of saponins.

The same authors have already published an abstract on the same topic: “Methods of Obtaining Extracts from Hedera helix L. Leaves and Evaluation of the Total Saponins Content” in Chem. Proc. 2022, 7, 56. https://doi.org/10.3390/chemproc2022007056. Some patents are already available (PI 0414535-6 BI) by other inventors.

In scientific manuscripts, the name of plants must be written as binomial scientific name and not using the common name. Thus, ‘Hedera helix extracts’ should be used, not ivy extracts.

Response: Thank you for your useful comments and suggestions on the manuscript, which are very helpful to our article. The word ”ivy” was changed with ”Hedera helix” throughout the manuscript.

Critical points

-          The binomial scientific name of plants must always be written in italics, the first name with the first letter uppercase and then lowercase as the second term, such as Hedera helix

Response: Thank you for the suggestion. The binomial scientific name of the plant material used in this research was modified throughout the manuscript, in accordance with your suggestion.

-           The last sentence of Abstract (L.22-24) is not clear; What is the meaning? The sentence should be modified to clear the sense of biological experiments.

Response: Thank you for your valuable comment. For a better understanding, the last paragraph of the Abstract was modified as follow (modified at page 1, lines 26-29):

“Until 200 µg/mL, the studied extracts were cytocompatible with L929 fibroblasts cell line at 48h of treatment. These in vitro cell culture results provide useful information for further applications of Hedera helix extracts in pharmaceutical field.”

-          The authors should improve the introduction and, in general, the text. It is very nonspecific... A description of other methods of extraction published in the literature must be reported.

Response: Thank you for the useful comment. The introduction was improved by incorporating, in this section, a description of other extraction methods for saponins. Thus, the following paragraph was inserted at page 2, lines 50-73:   

“To overcome these drawbacks, various non-conventional methods to extract saponins were successfully employed. Li et al. reported that the extraction of saponins from Aralia taibaiensis root bark by ultrasounds (at a temperature of 61 °C, an ethanol concentration of 73%, a solid to liquid ratio of 16 g/mL, and an extraction time of 34 min) led to better yields as compared with the conventional hot reflux extraction [12]. Akbari et al. studied the influence of different parameters on the microwave assisted extraction of saponins from fenugreek seed, concluding that the concentration of ethanol was the most significant factor, while temperature exhibited the least contribution on the total saponins content [13]. Supercritical fluid extraction of saponins from Medicago sativa was successfully employed by Kielbasa et al. Moreover, they developed a method for qualitative and quantitative analysis of saponins which involved acid hydrolysis with concentrated hydrochloric acid of the extracts and purification by supercritical fluid extraction with C18 [14]. Another study presents the subcritical-water extraction of triterpene saponins from Orostachys japonicus. Different temperatures and extraction times were investigated, concluding that the triterpene saponins can be observed only at 220 °C and 15 min of extraction [15]. Shahi et al. studied the pulsed electric field extraction of saponins from Chubak root and the possibility to use the extracts as emulsion and foam system for ice cream. They revealed that increasing the pulse number from 10 to 65 and the voltage from 1 to 4kV/cm lead to a better yield. This resulted in an increase of foaming ability and emulsion stability [16]. The enzymatic assisted extraction is another interesting technique being employed successfully to extract saponins from Sapindus mukorossi. Chen et al. reported a total saponin content of approximatively 116 mg/mL for 3 h of extraction, a plant material to solvent ratio of 1:5, and a cellulase concentration of 0.55% [17].”

-          Results: line 108. Are the authors sure about ‘by the conventional method (34 mgDE / gDM)’? From figure 1a, the value appears higher… > 35 mg DE/g DM.

Response: We are very sorry for the mistake and thank you very much for the comment. The correct value is the one in the chart. Thus, it was modified in the text in accordance with the one shown in the chart (please, see page 4, line 142). 

-          Results L.137-251. The description and discussion of the experimental conditions used for saponin extraction are too extensive and repetitive. This part must be written in shorter form.

Response: Thank you for the valuable suggestion. The subchapter 2.1 has been written in a shorter form, in accordance with your suggestion. Moreover, an ANOVA statistical analysis of the results was performed and incorporated in the revised manuscript. Please, see pages 3-12, lines 96-314.

-          Table 1 What is the statistical significance of the reported values? There is probably no statistical difference among the results obtained with 100 and 200 µg/mL. A statistical test must be performed. Furthermore, the data in the table are different from those reported in the text.

Response: Thank you for the useful comment. For the revised manuscript the statistical analysis of the cytotoxicity results obtained by in vitro MTT assay was performed using the ANOVA test, instead of Student’s test. According to ANOVA test of the reported values for the tested extracts, p-value is less than 0.05. These proves that there is a statistical difference among the results obtained with 100 and 200 µg/mL.

Regarding the data presented in Table 1 and those reported in the text, there was a typing mistake for which we are very sorry. The correct value is the one shown in Table 1. Thus, we modified the phrase which refers to the values presented in Table 1, as follow (please, see page 15, lines 370-372):

“At a concentration of 400 µg/mL, the total DNA content ranged from 50.84 % for hederacoside C to 70.46 % for the 80 % ethanolic extract and to 75.69 % for the 96 % ethanolic extract.”

-          The conclusion must be improved. From the reported results, there is no significant variation of cell proliferation and therefore DNA content up to 200 µg/ml of extracts, while 400 µg/mL are cytotoxic. Overall, one-way analysis of variance (ANOVA) and an appropriate post hoc test must be done.

Response: Thank you for your useful comments and suggestions on the manuscript. We performed an ANOVA test of the results. Also, the conclusion was improved. Please, see page 18, lines 502-510:

“The cytotoxicity assay of the samples indicated that until 200 μg/mL both ethanolic extracts were biocompatible on L929 fibroblast cells. The DNA content of fibroblasts treated with the extract samples and with the saponin standard were in accordance with the cell viability results determined by MTT assay. Both methods showed that the viability values of cells cultivated in the presence of studied extracts were dependent on the tested samples concentration. Additional in vitro and in vivo studies are required to characterize Hedera helix extracts in order to be used in pharmacological applications.”

All the results were subjected to one-way analysis of variance (ANOVA) and to the post hoc ?-test in order to detect the significant statistical differences between the averages of the TSC of two or more independent groups. The statistical evaluation was added to each subchapter.

-          What is the relation between the L929 fibroblast line and the immune stimulating activity of Hedera helix extracts? Please, this point must to be explained and/or the objective of the section in the manuscript “UAE of Active Principles from Ivy Leaves - Selection of Extraction Conditions in Correlation 70 with the Immunostimulatory Compounds from the Extracts” must be changed.

Response: Thank you for the comment. There is no direct relation between the L929 fibroblast cell line and the immune stimulating activity of Hedera helix extracts.  In our study, we assessed the biocompatibility of Hedera helix extracts on fibroblast cells. This is a primordial condition which must be fulfilled by the tested samples to manifest an immune stimulating activity. However, the title of the subsection 2.1 was modified as follow (please, see page 3, line 97-98):

“Influence of UAE parameters on the extraction efficiency”

-  There are several typographical errors... such as ‘in vitro’ should be written in italics, etc.

 In the pdf of the manuscript, some critical points/words are highlighted in  orange colour.

Response: Thank you for your useful suggestion. The word ”in vitro” was modified throughout the manuscript, in accordance with your suggestion. Also, the highlighted words/critical points in the revised manuscript were modified in accordance with your suggestion.

Reviewer 3 Report

In the manuscript subjected to revision Authors examined several factors influencing efficacy in saponin extraction from leaves of Hedrea helix by sonication.

General comments:

1)     English language correction is required.

2)     There is no statistical evaluation of data presented in the text.

3)     The legends on figures should be place above or below the charts, in current form they are dominating.

4)     Latin names of plants should be written using Italic font.

5)     Student’s t-test is not adequate to assess the data obtained by Authors. Anova should be used.

6)     It is not clear what for the DNA content was investigated – this part is irrelevant to the study and should be removed. There is no question posed that is addressed by this investigation.

Specific comments:

1)     Abstract - requires considerable improvement:

Line 17: what was the criteria to discern "best samples"?

Line 18: plant? – means plant material?; “ best conditions:” – means the most efficient in extraction?

Line 20: “20/1 (v/w)” – please use “:” to express ratio, also in the main text; “80% ethanol concentration in water” - just 80% ethanol;

Line 21: what for DNA content was investigated?

Lines 22 -25: the cytotoxicity was investigated so why the stimulation of proliferation is reported?

2)     Results and Discussion - requires considerable improvement:

Line 70: this title is irrelevant, it should be directly connected with the results presented and discuss below e.g. The effect of UAE parameters on extraction efficacy;

Figure 1: there is no statistical significance indicated; why on (a) also US power is presented but on (b) not?

Figure 2, 3, 4, 5, 6, 7: there is no statistical significance indicated;

Lines 107-108: unclear;

Lines 109-111: “These….slightly.” - such statemen should be proved by statistical assessment;

Line 129: why therefore?

Lines 147-148: “At 50 °C, the use of acoustics leads to a higher amount of extracted saponins.” - is this general statement related to results of this study?

Lines 160-161: “These results can be explained by the degradation of the thermolabile components.” - for which temperature 40 or 50 degrees? both?

Line 179: “total extracted saponins” – total content?

Lines 180-181: “Among the extracted saponins, hederacoside C and hederin have immunostimulatory properties.” - irrelevant information here;

Lines 181-185: “According to the HPLC analysis, these components are extracted from the first minutes of extraction (see Figure 3b). This represents an advantage considering that the scaling up of the extraction process is more economical for shorter times. A slight increase in the concentration of active components can be observed after 100 min. However, after 60 min the contents remain almost constant.” - unclear, and ther is no statistical assessment;

Lines 185-186: “This could be explained by a slight lability of the key compounds at long extraction times.” - or that extraction was completed;

Lines 192-195: “As shown in Figure 4a, a higher TSC value is achieved by increasing the plant to solvent ratio from 1/10 to 1/20. Thus, at a plant to solvent ratio of 1:10 (w/v), the TSC was 46.95 mgDE/gDM. The extraction at a ratio of 1/20 lead to similar saponin content (51.27 mgDE/gDM).” - this is not proved till statistical assessment of data;

Line 248: why consequent?

Author Response

In the manuscript subjected to revision Authors examined several factors influencing efficacy in saponin extraction from leaves of Hedrea helix by sonication.

General comments:

  • English language correction is required.

Response: We are very sorry for the language problems in the original manuscript. We have modified and polished the revision.

  • There is no statistical evaluation of data presented in the text.

Response: Thank you for your useful comments and suggestions on the manuscript. All the results were subjected to one-way analysis of variance (ANOVA) and the post hoc ?-test in order to detect the significant statistical differences between the averages of the TSC of two or more independent groups. The statistical evaluation was added to each subchapter.

  • The legends on figures should be place above or below the charts, in current form they are dominating.

Response: Thank you for the valuable suggestion. The legends on figures were modified in accordance with your suggestion (please, see figures 1-7).

  • Latin names of plants should be written using Italic font.

Response: We are very sorry for the mistake and thank you very much for the comment. The Latin name of the plant was modified throughout the manuscript, in accordance with your suggestion.

  • Student’s t-test is not adequate to assess the data obtained by Authors. Anova should be used.

Response: Thank you for the useful suggestion. For the revised manuscript the statistical analysis of the cytotoxicity results obtained by in vitro MTT assay was performed using the one-way ANOVA analysis, instead of Student’s test.  

  • It is not clear what for the DNA content was investigated – this part is irrelevant to the study and should be removed. There is no question posed that is addressed by this investigation.

Response: Thank you for your affirmation. The DNA content was also investigated to highlight the biocompatibility of the extracts by two different methods (MTT assay and DNA contained).

Specific comments:

1)     Abstract - requires considerable improvement:

Line 17: what was the criteria to discern "best samples"?

Response: Thank you for your useful comment. By ”best samples” we meant that we selected the extracts richest in saponins. For a better understanding, the phrase was modified as follow (please, see page 1, line 15-17):

“The aim of this study was to establish the best ultrasound assisted extraction (UAE) conditions of saponins from Hedera helix L. leaves and to evaluate the in vitro biocompatibility of the extracts richest in saponins.”

Line 18: plant? – means plant material?; “ best conditions:” – means the most efficient in extraction?

Response: Thank you for your comments. By ”plant” we meant ”plant material”, thus we changed the ”plant” word in ”plant material” throughout the manuscript. By ”best conditions” we meant those extraction parameters which lead to the most efficient extraction. For a better understanding we modified the phrase as follow (please, see page 1, lines 18-21):

” The most efficient extraction conditions were a temperature of 50 °C, an ultrasound amplitude of 40%, an extraction time of 60 min, a plant material to solvent ratio of 1:20 (w:v), and 80% ethanol as solvent.”

Line 20: “20/1 (v/w)” – please use “:” to express ratio, also in the main text; “80% ethanol concentration in water” - just 80% ethanol;

Response: Thank you for the valuable suggestions. We modified the manuscript in accordance with your suggestions: the ratios were expressed with “:” instead of “/” and the expression “x% ethanol concentration in water” was modified in  “x% ethanol” (where “x” is the concentrations used to perform the experiments – 0, 20, 40, 60, 80, and 96%).

Line 21: what for DNA content was investigated?

Response: Thank you for the comment. The DNA content was also investigated to highlight the biocompatibility of the extracts by two different methods (MTT assay and DNA contained).

Lines 22 -25: the cytotoxicity was investigated so why the stimulation of proliferation is reported?

Response: Thank you for the valuable comment. There was a wording mistake. We replaced the phrase “stimulation of the cell proliferation" with "cytocompatibility” throughout the manuscript.

2)     Results and Discussion - requires considerable improvement:

Line 70: this title is irrelevant, it should be directly connected with the results presented and discuss below e.g. The effect of UAE parameters on extraction efficacy;

Response: Thank you for the useful suggestion. The title was modified as follow (please, see page 3, lines 96-98):

“Influence of UAE parameters on the extraction efficiency”

Figure 1: there is no statistical significance indicated; why on (a) also US power is presented but on (b) not?

Response: Thank you for the useful suggestion. The US power values corresponding to each US amplitude has been introduced in Figure 1b. Also, the statistical significance was evaluated and introduced both in figure description and text, as follow (please, see page 4 lines 139-140 and lines 146-148):

“The one-way ANOVA analysis demonstrated that by increasing the ultrasonic power from 0 to 27.9 W the TSC increases significantly (p < 0.05). If the power is increased to 58.9 W, the TSC variation is not significant.”

Figure 2, 3, 4, 5, 6, 7: there is no statistical significance indicated;

Response: Thank you for the valuable comment. The statistical significance was evaluated and introduced both in figures descriptions and text.

Lines 107-108: unclear;

Response: Thank you for noticing. There was a written mistake of the TSC value. The correct value is the one in the chart. Thus, it was modified in text in accordance with the one shown in the chart (please, see page 4, line 141-142):

“As shown in Figure 1a, the lowest content is obtained for the extraction of saponins by conventional method (38 mg DE/g DM).”   

Lines 109-111: “These….slightly.” - such statemen should be proved by statistical assessment;

Response: Thank you for the valuable comment. The statistical significance was evaluated and introduced in text as follow (please, see page 4, lines 146-148):

“The one-way ANOVA analysis demonstrated that by increasing the ultrasonic power from 0 to 27.9 W the TSC increase significantly (p < 0.05). If the power is increased to 58.9 W, the TSC variation is not significant”

Line 129: why therefore?

Response: Thank you very much for noticing. There was a writing mistake. We deleted the word “therefore”. Please, see page 5, line 177.

Lines 147-148: “At 50 °C, the use of acoustics leads to a higher amount of extracted saponins.” - is this general statement related to results of this study?

Response: Thank you for your comment. The affirmation “At 50 °C, the use of acoustics leads to a higher amount of extracted saponins.” was completed after statistical analysis as follow (please, see page 5, lines 181-184):

“At a temperature of 50 ℃, the use of acoustics leads to a higher amount of extracted saponins (50.27 mg DE/g DM). The increase of temperature leads to a significant increase of TSC (p < 0.05) for all extraction times studied.”

Lines 160-161: “These results can be explained by the degradation of the thermolabile components.” - for which temperature 40 or 50 degrees? both?

Response: Thank you for the comment.  The sentence was completed as follow (please, see page 7, lines 213-214):

“These results can be explained by the degradation of the thermolabile components at both 40 and 50 ℃.”

Line 179: “total extracted saponins” – total content?

Response: Thank you very much for noticing. There was a wording mistake. By total extracted saponins we meant all three saponins identified by HPLC. For a better understanding, we modified the phrase as follow (please, see page 8, line 236-237):

“In Figure 3 it can be noticed that all three saponins identified by HPLC are directly proportional with extraction time and temperature.”

Lines 180-181: “Among the extracted saponins, hederacoside C and hederin have immunostimulatory properties.” - irrelevant information here;

Response: Thank you for the valuable comment. This sentence has been deleted.

Lines 181-185: “According to the HPLC analysis, these components are extracted from the first minutes of extraction (see Figure 3b). This represents an advantage considering that the scaling up of the extraction process is more economical for shorter times. A slight increase in the concentration of active components can be observed after 100 min. However, after 60 min the contents remain almost constant.” - unclear, and ther is no statistical assessment;

Response: Thank you for the valuable comment. The paragraph was modified as follow (please, see page 8, lines 242-244):

“According to the HPLC analysis, these components are extracted from the first minutes of extraction (see Figure 3b). Hederacoside C content increases up to 100 min, while α-hederin and hederagenin amounts remain almost constant.”

Lines 185-186: “This could be explained by a slight lability of the key compounds at long extraction times.” - or that extraction was completed;

 Response: Thank you for your suggestion. The sentence was modified, in accordance with your suggestion, as follow (please, see page 8, lines 244-246):

“This could be explained by a slight lability of the key compounds at long extraction times or that the extraction was completed.”

Lines 192-195: “As shown in Figure 4a, a higher TSC value is achieved by increasing the plant to solvent ratio from 1/10 to 1/20. Thus, at a plant to solvent ratio of 1:10 (w/v), the TSC was 46.95 mg DE/g DM. The extraction at a ratio of 1/20 lead to similar saponin content (51.27 mgDE/gDM).” - this is not proved till statistical assessment of data;

Response: Thank you for the valuable comment. The statistical significance was evaluated and introduced in text as follow (please, see page 9, lines 251-257):

“The ANOVA analysis along with multiple comparison post hoc tests (n = 3) showed that TSC increases significantly (p < 0.05) by increasing the plant material to solvent ratio from 1:10 to 1:20. As shown in Figure 4a, at a plant material to solvent ratio of 1:10 (w/v), the TSC was 46.95 mg DE/g DM, while at a ratio of 1:20 the TSC was 51.27 mg DE/g DM.”

Line 248: why consequent?

Response: Thank you very much for noticing. There was a wording mistake. For a better understanding, the sentence was modified as follow (please, see page 12, lines 316-319):

“To explore the potential application of the 96% and 80% ethanolic extracts in the pharmaceutical field, their in vitro biocompatibility was evaluated.”

Round 2

Reviewer 2 Report

The manuscript entitled ‘Ultrasound Assisted Extraction of Saponins from Hedera helix L. and in vitro Biocompatibility Evaluation of the Extracts’ is a manuscript on extraction methods useful to obtain Hedera helix extracts of leaves with high amount of saponins.

The manuscript has been revised to improve it, but some minor critical points need to be further modified.

Minor Critical Points

1.      Abstract line 20 Usually, the statistical method applied is not reported in the abstract.

2.      Introduction line 74_ UAE_ The acronym should be explaining not only in the abstract, but also for the first time in the manuscript. All acronyms should be explained in the manuscript.

3.      Introduction line 81 Take ‘L.’ from the binomial name; it should be reported only in the first appearance in the manuscript or, alternatively, all times.

Furthermore, a space was lost in ‘hederasaponins’.

4.      Results and Discussion Line 91 Eliminate ‘with less energy consumption’ because it is not pertinent.

5.      Figure 1, 2, 3 etc. The authors reported in the caption: ‘Data were analyzed using one-way ANOVA (p < 0.05) and multiple comparison post hoc t-tests (n = 6)’. But in the figures there are no symbols of statistical tests... It must be modified or explained. Furthermore, I suggest that A and B should be added to each figure in a different position, as reported in the attached PDF of the manuscript.

Statistical significance should be appropriately reported also in graphs and tables.

6.      Lines 184 and 185 The sentence ‘In Figure 3 it can be seen that the three saponins identified by HPLC are directly proportional to extraction time and temperature’ does not correct in relation to the figures.

7.      Line 256 It is not a negative control, but only a control (not a treatment, of any type).

Author Response

  1. Abstract line 20 Usually, the statistical method applied is not reported in the abstract.

Response: Thank you for your useful comment. The phrase referring to the statistical method was deleted. Please, see page 1, lines 15-28.

  1. Introduction line 74_ UAE_ The acronym should be explaining not only in the abstract, but also for the first time in the manuscript. All acronyms should be explained in the manuscript.

Response: Thank you for the valuable comment. The acronym ”UAE” was explained also in its first appearing in the manuscript. Please, see page 2, line 73.

  1. Introduction line 81 Take ‘L.’ from the binomial name; it should be reported only in the first appearance in the manuscript or, alternatively, all times.

Furthermore, a space was lost in ‘hederasaponins’.

Response: Thank you for the valuable comment. The ”L.” from the binomial name was deleted, in accordance with your suggestion, throughout the manuscript.

For a better understanding the word “hederasaponins” was replaced with “saponins”. Thus, the phrase was modified as follow (please, see page 2, lines 85-87):

“The purpose of this study was to evaluate the process parameters of UAE of saponins from Hedera helix in order to establish an extraction protocol for the major saponins found in Hedera helix leaves.”  

  1. Results and Discussion Line 91 Eliminate ‘with less energy consumption’ because it is not pertinent.

Response: Thank you for the suggestion. The phrase ”with less energy consumption” was deleted. Please, see page 3, line 107.

  1. Figure 1, 2, 3 etc. The authors reported in the caption: ‘Data were analyzed using one-way ANOVA (p < 0.05) and multiple comparison post hoc t-tests (n = 6)’. But in the figures there are no symbols of statistical tests... It must be modified or explained. Furthermore, I suggest that A and B should be added to each figure in a different position, as reported in the attached PDF of the manuscript.

Statistical significance should be appropriately reported also in graphs and tables.

Response: Thank you for the suggestion. Letters for the statistical tests were inserted in the figures.

The ”A” and ”B” of the figures were modified in accordance with your suggestion. Please, see figures 1, 3, 4, and 5.

  1. Lines 184 and 185 The sentence ‘In Figure 3 it can be seen that the three saponins identified by HPLC are directly proportional to extraction time and temperature’ does not correct in relation to the figures.

Response: Thank you for the useful comment. For a better understanding, the phrase was modified as follow (please, see page 8, lines 236-238):

”In Figure 3 it can be noticed that hederacoside C is directly proportional to extraction time and temperature. However, α-hederin and hederagenin contents remain almost constant regardless of these two parameters.”

  1. Line 256 It is not a negative control, but only a control (not a treatment, of any type).

Response: Thank you for the valuable comment. The word ”negative” was deleted throughout the manuscript.

Reviewer 3 Report

The following aspect should be still addressed:

the statistical significance should be also presented on figures, not only in the text e.g. Figure 1a., Figure 4a, Figure 5a etc.

Author Response

The statistical significance should be also presented on figures, not only in the text e.g. Figure 1a., Figure 4a, Figure 5a etc.

Response: Thank you for your useful comments and suggestions on the manuscript. We presented the statistical significance also on figures and tables.